# Screening the Significant Hub Genes by Comparing Tumor Cells, Normoxic and Hypoxic Glioblastoma Stem-like Cell Lines Using Co-Expression Analysis in Glioblastoma

**DOI:** 10.3390/genes13030518

**Published:** 2022-03-15

**Authors:** Emine Güven, Muhammad Afzal, Imran Kazmi

**Affiliations:** 1Department of Biomedical Engineering, Düzce University, Düzce 81620, Turkey; emine.guven33@gmail.com; 2Department of Pharmacology, College of Pharmacy, Jouf University, Sakaka 72341, Al Jouf, Saudi Arabia; 3Department of Biochemistry, Faculty of Science, King Abdulaziz University, Jeddah 21589, Makkah, Saudi Arabia; ikazmi@kau.edu.sa

**Keywords:** biomarker, differentially expressed genes, co-expression, VEGF signaling pathway, gene ontology pathway enrichment, glioblastoma multiforme

## Abstract

Glioblastoma multiforme (GBM) is categorized by rapid malignant cellular growth in the central nervous system (CNS) tumors. It is one of the most prevailing primary brain tumors, particularly in human male adults. Even though the combination therapy comprises surgery, chemotherapy, and adjuvant therapies, the survival rate is on average 14.6 months. Glioma stem cells (GSCs) have key roles in tumorigenesis, progression, and counteracting chemotherapy and radiotherapy. In our study, firstly, the gene expression dataset GSE45117 was retrieved and differentially expressed genes (DEGs) were spotted. The co-expression network analysis was employed on DEGs to find the significant modules. The most significant module resulting from co-expression analysis was the turquoise module. The turquoise module related to the tumor cells, hypoxia, normoxic treatments of glioblastoma tumor (GBT), and GSCs were screened. Sixty-one common genes in the turquoise module were selected generated through the co-expression analysis and protein–protein interaction (PPI) network. Moreover, the GO and KEGG pathway enrichment results were studied. Twenty common hub genes were screened by the NetworkAnalyst web instrument constructed on the PPI network through the STRING database. After survival analysis via the Kaplan–Meier (KM) plotter from The Cancer Genome Atlas (TCGA) database, we identified the five most significant hub genes strongly related to the progression of GBM. We further observed these five most significant hub genes also up-regulated in another GBM gene expression dataset. The protein–protein interaction (PPI) network of the turquoise module genes was constructed and a KEGG pathway enrichments study of the turquoise module genes was performed. The VEGF signaling pathway was emphasized because of the strong link with GBM. A gene–disease association network was further constructed to demonstrate the information of the progression of GBM and other related brain neoplasms. All hub genes assessed through this study would be potential markers for the prognosis and diagnosis of GBM.

## 1. Introduction

One of the most prevailing and highly deadly heterogeneous forms of brain tumors is glioblastoma multiforme (GBM) or grade-IV glioma [1]. The diagnosis of GBM patients is very challenging, and the patient survival rate is 12–15 months even with combinational therapies [2]. The low efficiency of all therapeutic methods including surgery, chemotherapy, and radiotherapy [3] demands pointed to new therapeutic targets for GBM in recent years.

GBM is an extremely heterogeneous tumor at the pathological and cellular level [4,5]. Gene expression and cell proliferation levels also highly differ in GBM. Glioma stem cells (GSCs) take a central position regarding tumor formation of lower-grade gliomas and glioblastoma multiforme. GSCs have important characteristics including self-renewal ability, tumor initiation, progression ability, and resistance to GBM therapies. Several important roles of GSCs in GBM make GSCs new therapeutic targets [6,7]. As the neoplastic cells emerge as immune cells, cancer can be observable since the tumor cells have an extensive clonogenic latent sort called cancer stem cells. In the wet lab conditions, glioblastoma stem-like cells efficiently disseminate in the media after being insulated from newly resected human GBM [8].

The metabolic relation between the glycolysis and pentose phosphate pathway has been discovered in GSCs [9]. Neoplastic cells consume glycolysis energy for uncontrolled cell growth and further division. The study further demonstrated that FLK-1 carries a key position in the development of Vasculogenic Mimicry (VM) in glioblastoma multiforme. They reported the clinical characteristics of SOX2 and original outcomes that may offer fresh medical purposes for SOX2 as a prognostic biomarker.

Hence, a restructured knowledge of GBM is essential, and direct machinery is crucial for modified and curable therapies to increase patient survival rates. The multi-gene expression profiling of diseased samples was driven by high-throughput sequencing technology publicly available through the GEO database [10]. Even though just a small fraction of these datasets have been taken and analyzed, additional aspects of the mechanism of rapid expansion and resistance to treatments of glioblastoma tumor should be highlighted. The gene expression microarray dataset GSE45117 [9] is re-evaluated and employed to propose useful results for additional investigation in silico.

This study aims to suggest a treatment to handle them to block the rapid progress by associating clinical data with a molecular mechanism. Currently, there are still no permanent therapeutic alternatives available for GBM. Furthermore, patients who are diagnosed with glioblastoma tumors has very low survival rates. Screening of pathways and proteins involved in chemotherapeutic resistance identification use genomic and proteomic analyses [11]. For instance, changes in the Interleukins protein family expression and related conditions in GBM progression and growth require extensive investigation [12].

The computational pipeline detects the genetic markers for tumor differentiation by determining discrepancies in expression levels of glioblastoma tumor cells, stem-like cells, and cell lines. The investigation of hub nodes between pairwise samples by treatments of a significant co-expression module of DEGs resulted in the construction of co-expression networks. The current project is intended to reveal the biological, cellular, and functional pathways and linked genetic mechanisms of GB tumor in the most significant module.

The empirical research [13,14] so far has only focused on screening the significant genes. This analysis further presented the study of the DEGs utilizing WGCNA. Moreover, GO and KEGG pathway studies were reported concerning the biological process, cellular component, and molecular function of the pathways of the common hub genes. Moreover, a PPI network was built, and the related signaling pathways were studied to identify most significant hub genes of DEGs in the GSE45117 dataset. The gene expression of the GSE124145 dataset was further studied for the verification of the upregulated expression of the most significant hub genes.

## 2. Materials and Methods

### 2.1. The Gene Expression Dataset

Microarray data for human glioblastoma and glioma stem-like cells were retrieved from the GEO database of NIH by typing in the search box the word “glioma”. The GSE45117 gene expression dataset includes total RNAs from samples of glioblastoma tumor (GBT), normoxic glioblastoma stem-like cell lines (GSN), normoxic glioblastoma stem-like cell line, exposed to hypoxia for 48 h (GSN_H), hypoxic glioblastoma stem-like cell line (GSH), and hypoxic glioblastoma stem-like cell line, exposed to normoxia for 48 h (GSH_N).

The GEOquery package in Bioconductor is used to analyze the GSE45117 dataset [15]. The list of other packages is Biobase, biomaRT, and gplots of R studio [15,16,17]. The Benjamini–Hochberg technique is used to correct multiple testing and calculate the adjusted *p*-value to avoid Type I errors. A hypergeometric model was performed for both the down and up-regulated DEGs in GO enrichment in categories and KEGG pathway studies [18,19]. Moreover, adjusting the statistical tests locally is done by calculation of a false discovery rate (FDR) [20,21]. A workflow of the data analysis step by step is drawn in Figure 1.

### 2.2. Gene Expression Analysis

The study was done in R version 3.6.3. The GBM dataset with low quality and low reads was excluded, yet the rest of the expression set was transformed to a base-2 logarithmic scale. Moreover, gene expression levels were normalized by averaging the treatments before conducting analyses. A general assessment of statistical implementation can be obtained by clustering samples utilizing the correlation metric. Dendrograms based on the correlation metric are useful for identifying outlying samples [22]. Samples presenting atypical distribution of noisy intensities might be an extensive issue. This can be balanced by utilizing non-normalized data to create a box plot of the log intensities, before using absolute signal intensities, which warrants a more even representation of data [23,24].

### 2.3. Statistics and Differentially Expressed Genes

The dataset was retrieved utilizing the GEOquery package in Bioconductor [25]. The statistical significance of *p*-value < 0.05 and |log2(FC)| > 0.5 was set to determine DEGs between each treatment category using student *t*-test for additional review. The study used the heatmap.2 function in the gplots package to generate heatmap plots of DEGs [17,26]. Moreover, the expression values were normalized for each data point in each case of expression data using a log(FC) transformation as follows:$$n_{ij}=\log_{2}\frac{case_{ij}}{mean_{i}}$$
where $case_{ij}$ represents the expression value of the $gene_{i}$ of the *j*th case and $mean_{i}$ is the average expression value of the gene *i* in the control samples [27].

### 2.4. Weighted Co-Expression Analysis

The union of DEG sets with the corresponding expression values within the treatments was utilized to detect the scale-independent gene modules of co-expression and strongly linked genes generated by WGCNA [28,29]. The topological overlap matrix (TOM) was estimated by adjacency conversion and picked the number (1-TOM) as a measure of distance for detecting genes and modules via hierarchical clustering [30]. Moreover, the parameter blockSize set to 30 and TOMType were assigned to nothing. The soft threshold power β was fixed to 7, the lowermost power founded on the scale-independent topology to construct a weighted gene network.

### 2.5. The PPI Network

A most significant module was selected, and the common hub genes were outlined by treatment weights and module connectivity, which resulted from co-expression analysis. Furthermore, to pick hub seeds, all the candidate genes in the significant module were uploaded by their corresponding average gene expression values to the NetworkAnalyst platform to build the PPI network utilizing STRING Interactome [31]. Additionally, we utilized a zero-order network tool, particularly one that enables to keep hub proteins that interact with each other directly. The proteins were determined with the degree connectivity > 2 (total edges/total nodes) as the hub genes in the PPI network to implement the co-expression network [32].

### 2.6. GO and KEGG Enrichments of the Pathways

Before studying annotations of the DEGs in the most significant module, affy IDs were matched with corresponding gene symbols using the Biomart package [16]. Consequently, gene ontology annotations regarding BP, CC, and MF using DAVID 6.8 and KEGG pathways were studied [19,33]. All of the annotations and subsequent hub genes were cautiously evaluated and separated based on the features of their biological and molecular significances.

### 2.7. Common Hub Gene Survival Analysis

The common hub genes portion in the stemness of GBM was examined via the positively correlated genes in TCGA of the gene expression dataset utilizing the UALCAN database [34]. The gene expression levels of hub genes at significance (*p*-value < 0.05) are studied. For deep analysis and validation, survival analysis of GBM patients and the significance of survival effect is measured by log-rank test [35]. The GEPIA2 multiple gene comparison tool [36] was utilized to pair TCGA normal and GTEx data of most significant hub genes.

### 2.8. Validation of the Common Hub Genes

#### 2.8.1. Analysis of a Separate Glioma Gene Expression Dataset

This study retrieved another human glioblastoma and glioma stem cells from the NIH Gene Expression Omnibus (GEO) by typing in the search box the word “glioma” in the GEO database. The GSE124145 gene expression dataset [37] includes total RNAs from three human glioblastoma multiforme tissues (hGBM), six human glioma stem cells (GSCs), and three glioma cell lines from direct tumor resection of a 54-year-old female patient. The DEGs of the GSE124145 gene expression dataset were studied separately. In particular, the expression levels of the most significant hub DEGs from the GSE45117 were focused on to validate the significance of them referencing the GSE124145 dataset.

#### 2.8.2. The Gene-Disease Association Network of the Common Hub Genes

This study further constructed a gene-disease network to confirm the association of the common hub genes with GBM and related genetic diseases or disorders via NetworkAnalysist. The network association information was collected from the DisGeNET database [38], which is a broad platform combining knowledge on human disease-associated genes and variants.

## 3. Results

### 3.1. A Hierarchical Clustering of the Clinical Dataset

Figure 2A illustrates the GSE45117 dataset examining hierarchical clustering that is beneficial for picturing clusters or groups of samples and comparative adjacencies. Figure 2B demonstrates the boxplot of the GSE45117 gene expression dataset within each treatment. The expression values were used to confer a paired comparison of volcano plots and designed heatmaps (Appendix A).

### 3.2. DEGs of Paired Tumor Subtypes

Following data preprocessing and quality evaluation, this study revealed the expression values from the 12 samples in GSE45117. A set of 2447 DEGs (1133 down-regulated and 1313 up-regulated) in *GBT*, *GSN*, *GSN_H*, *GSH*, and *GSH_N* clinical features, under the threshold of *p*-value < 0.05 and |log_2_(FC)| > 0.5 (Table 1) were screened for the consequent analyses. The heatmaps and volcano plots of DEGs are provided in Appendix A.

### 3.3. Co-Expression Analysis

Employing the average linkage-clustering technique (Figure 3A) based on samples of DEGs of the GSE45117 dataset and to offer a scale-independent network, β = 7 (unscaled R^2^ = 0.90) power was chosen (Figure 3B). The significant five modules were observed (Figure 3C) using two techniques to assess the relationship among modules and progression of the GBM treatments. Module hierarchical clustering analysis (Figure 3C) revealed the turquoise module had a greater correlation with treatments than different modules (Figure 3D). Moreover, the turquoise module was positively correlated with GBT cells but was not correlated with GSH_N and GSH (white shading), and was negatively correlated with GSN_H and GSN (blue shading) (Figure 4). Hence, the analysis noted the turquoise module of progression in treatments as the most correlated based on the two methods.

### 3.4. The Hub Module: GO and KEGG Pathway Enrichments

In Table 2 and Figure 5, the BP, CC, and MF of the GO annotations on DAVID are listed for 61 common hub genes in the turquoise module at a significance level (*p*-value < 0. 05) (See Appendix A for the entire list).

This shows the significant enrichments of most candidate hub genes in BP terms are GO:0006955~immune response, GO:0006952~defense response, GO:0050900~leukocyte migration, GO:0006954~inflammatory response, and GO:0002682~regulation of immune system process. The significant enrichment GO terms in CC are revealed as GO:0005887~integral component of plasma membrane, GO:0031226~intrinsic component of plasma membrane, GO:0005615~extracellular space, GO:0044421~extracellular region part, and GO:0031988~membrane-bounded vesicle. Lastly, the significant enrichment of the hub genes in MF contains GO:0005102~receptor binding, GO:0032403~protein complex binding, GO:0005178~integrin binding, GO:0004872~receptor activity, and GO:0060089~molecular transducer activity. KEGG signaling pathway enrichment study reported that the hub genes were significantly enriched in hsa05150:staphylococcus aureus infection, hsa05144:Malaria, hsa04380:Osteoclast differentiation, hsa04064:NF-kappa B signaling pathway, and hsa05134:Legionellosis.

### 3.5. Screening Common Hub Genes and PPI Network

To achieve the analysis of protein–protein interactions in the turquoise module (1025 DEGs), the PPI network was built on the common hub genes list. Through the PPI network, 123 nodes and 219 edges were selected (Figure 6) A total of 61 seeds (proteins) which were highly connected with the turquoise module were screened as common hub genes as a result of PPI network construction.

Twenty primary nodes (proteins) with the greatest degrees in the gene expression of GSE45117 dataset common hub genes were picked and are listed in Table 3. These are *SRC*, *SYK*, *EGFR*, *LYN*, *RAC2*, *SORBS2*, *IL1R1*, *PLCG2*, *S100A8*, *CCL8*, *PIK3CG*, *RHOG*, *CD44*, *TLR4*, *RHOU*, *EGR1*, *DAB2*, *KDR*, *ITGB2*, *HCK*. Hub genes in Figure 6 following a gradual color change from green to light green and brown to light brown represent expression intensity. The size of nodes represents fold change (FC). The turquoise module was linked to the connectivity degree in the co-expression network. In Figure 6, a negative correlation in the brown nodes and a positive correlation in the green nodes can be seen.

The PPI network of common hub gene KEGG enrichment analysis picks the VEGF signaling pathway as the most significant (*p*-value < 0.05) pathway (Table 4). The VEGF signaling pathway demonstrates an important role in many cancers is one of the leading angiogenic regulator pathways involving GBM through hyperactivation. It is also of concern in various biomarkers of tumorigenic progression such as proliferation and survival.

### 3.6. Authentication of the Five Most Significant Hub Genes

The top 20 common hub genes (Table 4) of the co-expression network were confirmed at expression levels and overall survival (OS) in TCGA datasets. After survival analysis using the GSE45117 and TCGA dataset, as illustrated in Figure 7 and Table 5, we identified the five most significant hub genes (*IL1R*, *SORBS2*, *S100A8*, *CCL8*, *DAB2*) strongly linked to the progression of GBM. Figure 8A demonstrates a histogram of the five most significant hub gene expression levels by treatments (GBT, GSN, GSN_H, GSH, and GSH_N). The multiple gene comparison analysis of hub genes using TCGA normal and GTEx datasets is shown in Figure 8B.

All five hub genes have shown a decrease in expression with GSN_H and GSH_N treatments. A dramatic drop in the expression of *IL1R1* and *S100A8* with GSN_H and GSH_N treatments is further reported in Table 5. The relation to gene expression study outcomes shows details about relative expression levels of *IL1R1* and *S100A8* in normal versus GB tumor samples as demonstrated in Figure 8C,D.

### 3.7. Validation of Five Most Significant Hub Genes

The expression level of *IL1R1*, *SORBS2*, *S100A8*, *CCL8*, and *DAB2* was focused on GSE124145. The outcomes of the analysis to validate the significance are demonstrated in Table 6; it was noted that all of the most significant hub genes (*IL1R1*, *SORBS2*, *S100A8*, *CCL8*, and *DAB2*) were expressed significantly upregulated in both GSE124145 as well. We further constructed a gene–disease association network as shown in Figure 9. The most significant hub genes were observed in the intersection of glioblastoma, giant glioblastoma, status epilepticus, and head and neck Neoplasms.

## 4. Discussion

GBM is the most prevalent, destructive, and fatal brain cancer. Present treatment decisions including surgery, adjuvant therapy, and chemotherapy cannot fully treat the disease, because the tumor is highly defiant to these treatments. GSCs have the self-renewal capacity and are accountable for the tumor resistance in treating GBM.

This study identified the DEGs in the GBT–GSN, GBT–GSN_H, GBT–GSH, GBT–GSH_N, GSN-GSN_H, GSN-GSH, GSN-GSH_N, GSN_H-GSH, and GSN_H-GSH_N treatment groups. Prior to the identification of DEGs, normalization is done based on a method used by Chung and Lee (2021). In this study, we performed WGCNA for mutual DEGs derived from the GEO database and reconstructed gene co-expression networks. First, we applied the WGCNA approach to DEGs (Table 1) of the GSE45117 dataset to evaluate the gene expression profile differences including GSCs, and a human GBT sample. As a data analysis approach or a gene filtering (screening) technique to identify groups (modules) of favorably related proteins, the WGCNA is an R software for weighted co-expression study and can be utilized [28,29]. Subsequently, GO and KEGG pathway enrichments were implemented on the turquoise module (*p*-value < 0.05), which resulted in the most significant module (Table 2).

The GO pathway analysis is revealed in biological process (BP) terms GO:0006955~immune response, GO:0006952~defense response, GO:0050900~leukocyte migration, GO:0006954~inflammatory response, and GO:0002682~regulation of immune system process. The significant enrichment GO terms in the cellular component (CC) are revealed as GO:0005887~integral component of plasma membrane, GO:0031226~intrinsic component of plasma membrane, GO:0005615~extracellular space, GO:0044421~extracellular region part, and GO:0031988~membrane-bounded vesicle. Lastly, the significant enrichment of the hub genes in molecular function (MF) contains GO:0005102~receptor binding, GO:0032403~protein complex binding, GO:0005178~integrin binding, GO:0004872~receptor activity, and GO:0060089~molecular transducer activity. The KEGG signaling pathway study reported that the hub genes were significantly enriched in hsa05150:staphylococcus aureus infection, hsa05144:Malaria, hsa04380:Osteoclast differentiation, hsa04064:NF-kappa B signaling pathway, and hsa05134:Legionellosis (Table 2 and Figure 5).

Twenty primary nodes (Figure 6) with the top degrees in the gene expression levels of DEGs are presented in Table 3. These genes can be listed as *SRC*, *SYK*, *EGFR*, *LYN*, *RAC2*, *SORBS2*, *IL1R1*, *PLCG2*, *S100A8*, *CCL8*, *PIK3CG*, *RHOG*, *CD44*, *TLR4*, *RHOU*, *EGR1*, *DAB2*, *KDR*, *ITGB2*, and *HCK*, and so-called common hub genes (Table 4 and Figure 7). The VEGF signaling pathway was screened as the most significant (*p*-value < 0.05) pathway (Table 4). The VEGF signaling pathway demonstrates an important role in many cancers involving GBM through hyperactivation and is of concern in various biomarkers of tumorigenic progression such as proliferation and survival. Furthermore, the VEGF signaling pathway is one of the leading angiogenic regulator pathways in these tumors [39,40]. Other KEGG pathway enrichments of the top 20 hub genes have resulted in hsa05205:Proteoglycans in cancer, hsa04666:Fc gamma R-mediated phagocytosis, hsa04664:Fc epsilon RI signaling pathway, hsa04662:B cell receptor signaling pathway, and hsa04064:NF-kappa B signaling pathway.

The survival and expression analyses of the common hub genes pick the five most significant hub genes such as Interleukin 1 Receptor Type 1 (*IL1R1*), Sorbin and SH3 Domain Containing 2 (*SORBS2*), S100 calcium-binding protein A8 (*S100A8*), C-C Motif Chemokine Ligand 8 (*CCL8*), and DAB Adaptor Protein 2 (*DAB2*). The hub genes are powerfully connected with the development of GBM and they might be useful as potential therapeutic agents as shown in Figure 9. A dramatic drop in the expression of *IL1R1* and *S100A8* with GSN_H and GSH_N treatments is further reported in Table 5. The association to gene expression analysis results proposes details about comparative expression levels of *IL1R1* and *S100A8* in normal versus GB tumor as shown in Figure 8C,D. Interleukin-1 signaling is established as an appealing and key therapeutic target for the controlling of glioblastoma-related cerebral edemas [41]. The role of the IL-1 gene family in glioblastoma linked angiogenesis and tumor development has been reported in several studies [41,42,43]. Therapeutically, a knockdown of the IL-1R1 might evaluate inhibition of IL-1 signaling as a novel therapy for GBM [44,45]. In a recent study, *SORBS2* in TCGA GBM cohorts has been reported among other genes as possibly being linked with inferior consequences and PDE1C silencing down-regulated their expression [46], consequently proving to be promising concerning patient survival. Furthermore, WFDC1 is an instance of *SORBS2*-bound transcripts which is mediated by *SORBS2* and a key metastasis suppressor [47]. Previous research has reported that WFDC1 expression was considerably downregulated in mesenchymal cells in brain cancer [48,49].

Recently, *S100A8* has been reported as a prospective indicator with prognostic and diagnostic value in GBM [50]. Gielen et al. (2016) proposed that glioma patients have enlarged quantities of intracellular S100A8/9 compared with healthy controls. Glioma patients further have boosted S100A8/9 serum quantities correlated with amplified arginase bustle in serum. S100A8/9 can be expressed by various myloid cells and tumor cells in glioma, where it can promote tumor cell growth and migration [51].

In a recent laboratory investigation [52], the data uncovered that *CCL8* is a tumor-associated macrophage element that resolves penetration and GBM stemness and has resistance to therapies. Moreover, it is reported in another study that *CCL8* stimulates the development of tumor cells in the glioma microenvironment [52]. Our study also verified targeting *CCL8* offers a new prospect for GBM treatment.

The usual treatment protocol for GBM involves surgical removal of the tumor at a maximal and healthy level, radiation therapy, and temozolomide (TMZ) chemotherapy which is broadly used for silencing GBM. It is shown that the loss of DAB2IP (DOC-2/DAB2 interacting protein) is liable for TMZ resistance in GBM through autophagy-related protein 9b (ATG9B). In a fresh study that listed four subtypes of GBM, *DAB2* is reported as one of the fifteen selected genes that belong to the classical (CL) subtype. S100A4 was found in the CL subtype of GBM [53].

To validate our results we further analyzed another clinical GBM gene expression dataset of GSE124145. All these genes were significantly up-regulated in both of the GBM datasets. Thus, targeting these five most significant hub genes (*IL1R1*, *SORBS2*, *S100A8*, *CCL8*, and *DAB2*) may offer insightful strategies for GBM treatment. To confirm the association with GBM, we constructed a gene–disease association network as shown in Figure 9. While we validated the results in the TCGA dataset, the accuracy of the results requires molecular and cellular experiments. Thus, by utilizing a sequence of bioinformatics investigation, this current study demonstrated the five most significant hub genes which may be tangled in the diagnosis and prognosis and efficient concerning the characterization of GBM and treatment options. A limitation of this study is the lack of experimental validation to confirm our results. The essential pathways enriched in the candidate hub genes were cell migration, cell motility, localization of cell, locomotion, and leukocyte migration. These results would significantly offer to uncover the progression of GBM.

## 5. Conclusions

Here, we focused on one of the GBM gene expression datasets in the GEO database. In this study, *IL1R1*, *SORBS2*, *S100A8*, *CCL8*, and *DAB2* were filtered as the most significant hub genes for the prospective molecular, metabolism, functional studies in GBM. We further validated the five most significant hub genes’ significance level using a similar clinical GBM gene expression dataset. Moreover, a gene–disease association network was constructed to confirm the impact of these five hub genes in GBM. These hub genes can be offered to the candidate markers of future research for therapeutic targets in GBM. The rest of the analysis in this study would help to explore the causes of the gliomas, in particular GBM, underlying biological, cellular, and functional events.

## Figures and Tables

**Figure 1 genes-13-00518-f001:**
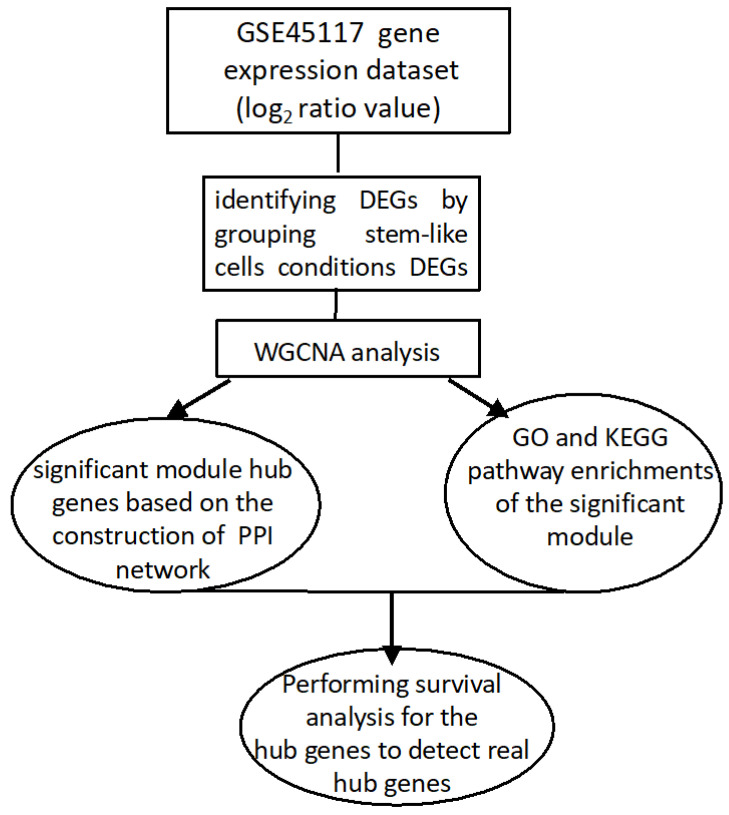
A design of the computation steps of the GSE45117 gene expression dataset.

**Figure 2 genes-13-00518-f002:**
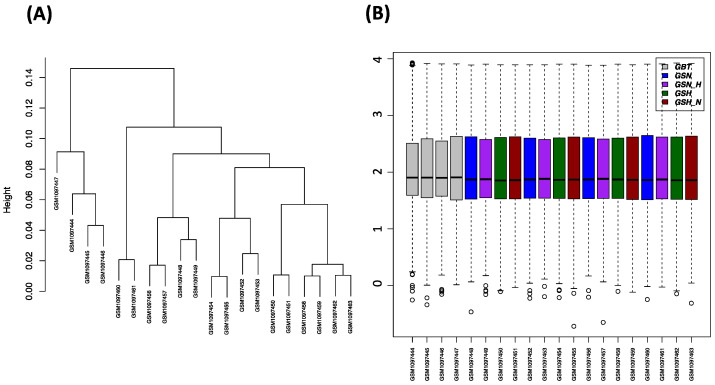
(**A**) A hierarchical clustering plot of gene expression dataset of expression values base-2 logarithmic value. (**B**) The boxplot of the GSE45117 gene expression dataset within each sample.

**Figure 3 genes-13-00518-f003:**
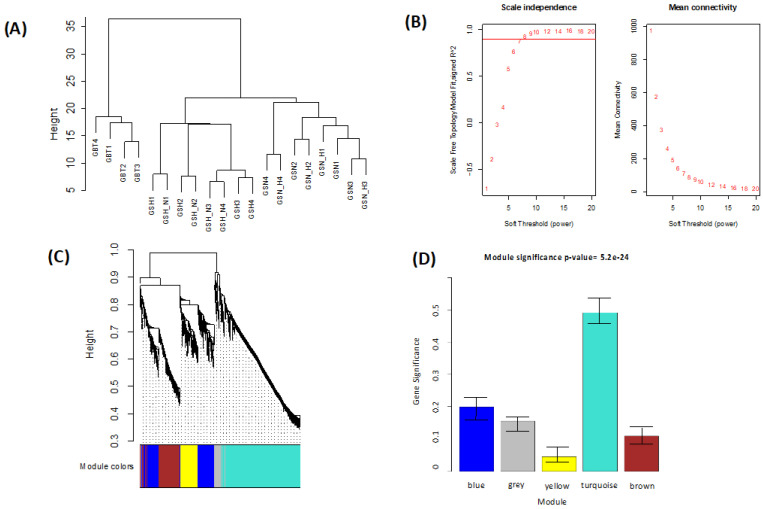
(**A**) Treatment samples clustering to identify outliers of GSE45117: Treatments dendrogram. (**B**) Identification of soft-thresholding power in the WGCNA. (left) The plot of the scale-independent fitting index for numerous soft-thresholding powers (β). (right) The plot of the average connectivity for numerous soft-thresholding powers. (**C**,**D**) Identification of modules linked to the tumor treatments of GSE45117. (**C**) Clustered dendrogram of all the DEGs with a dissimilarity measure (1-TOM). (**D**) Bars of the mean gene significance distribution and each module’s error on the corresponding bar linked to the treatments of GBM.

**Figure 4 genes-13-00518-f004:**
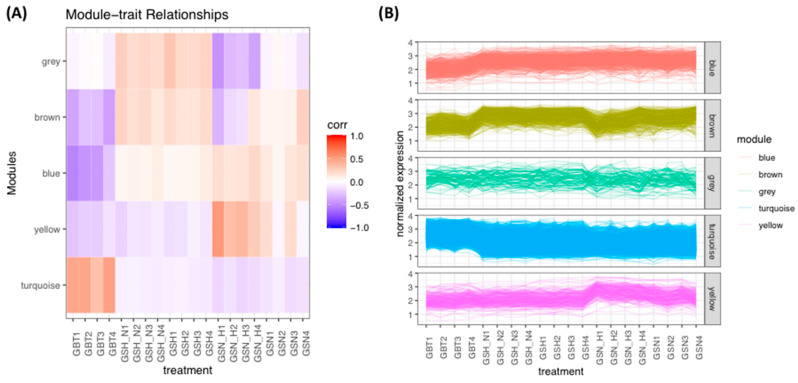
(**A**) The correlation heatmap by module eigengenes and the treatments in DEGs of the GSE45117. (**B**) The plot of normalized expression values of DEGs and the modules related to the treatments of the GBM dataset.

**Figure 5 genes-13-00518-f005:**
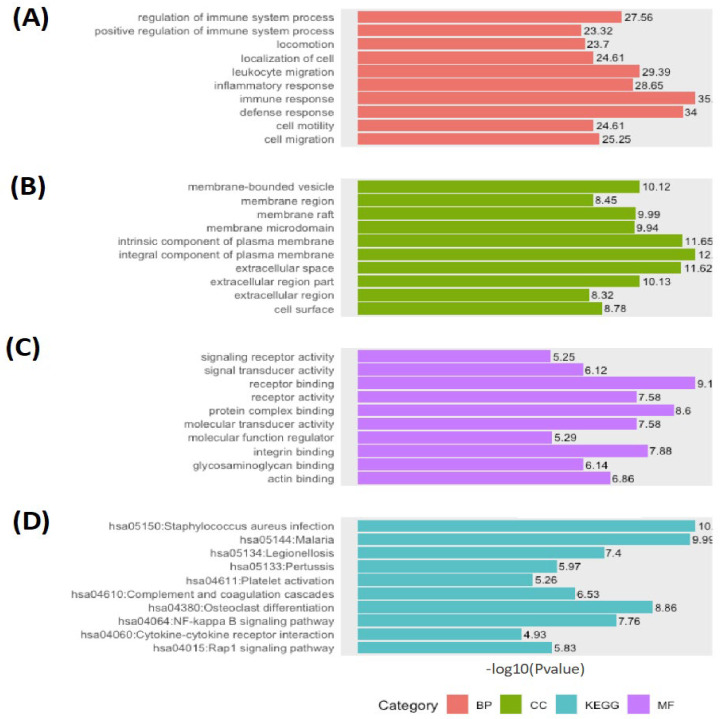
The plot illustrates the top GO annotations in terms (**A**) BP, (**B**) CC, and (**C**) MF of the turquoise module. (**D**) The top KEGG pathways of the most significant module (turquoise) are demonstrated.

**Figure 6 genes-13-00518-f006:**
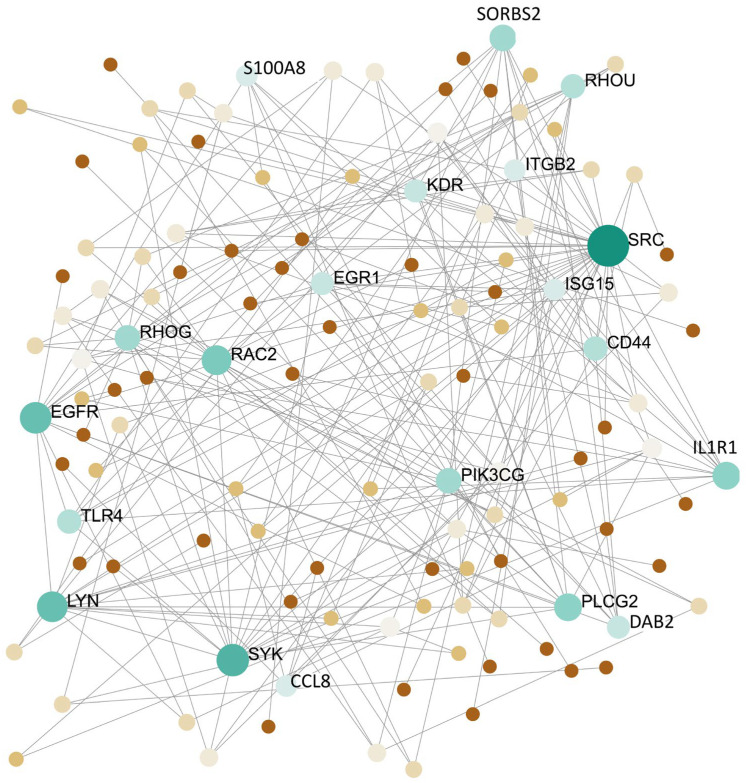
The protein–protein interactions in the turquoise module. The gradual color change from green to light green and brown to light brown represents expression intensity. The turquoise module was akin to the connectivity degree in the co-expression network, correlated negatively in the brown and correlated positively in the green nodes. The perimeters of the nodes were analogous to the fold change (FC).

**Figure 7 genes-13-00518-f007:**
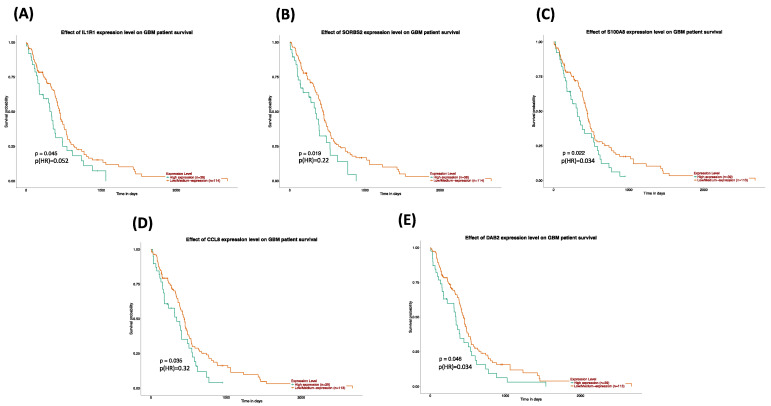
Survival analysis of Kaplan–Meier (KM) plots of the most significant hub genes in the TCGA dataset via the UALCAN. (**A**) *IL1R1*, (**B**) *SORBS2*, (**C**) *S100A8*, (**D**) *CCL8*, and (**E**) *DAB2*. Orange lines represent the low expression of the most significant hub genes, whereas green lines represent high expression.

**Figure 8 genes-13-00518-f008:**
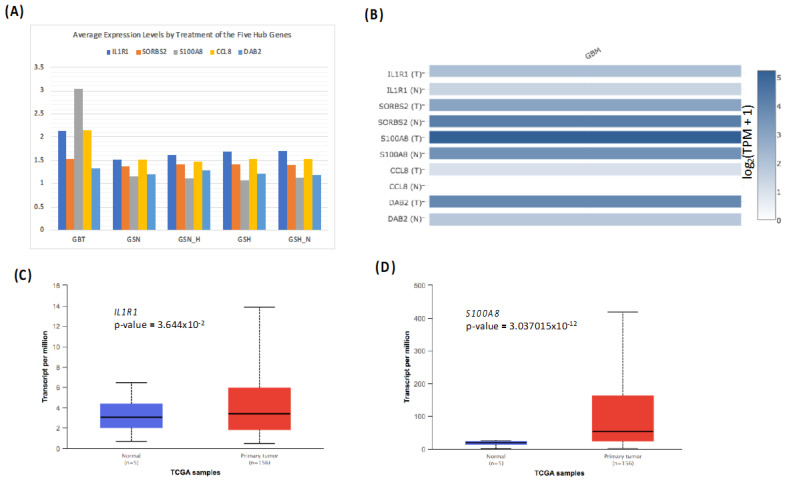
(**A**) Histogram of the five most significant hub gene expression levels by treatments. (**B**) The multiple gene comparison studies of most significant hub genes are plotted on TCGA normal and GTEx datasets. Boxplot showing relative expression of (**C**) *IL1R1* and (**D**) *S1000A8* in normal (*n* = 3) and GBM (*n* = 156) samples.

**Figure 9 genes-13-00518-f009:**
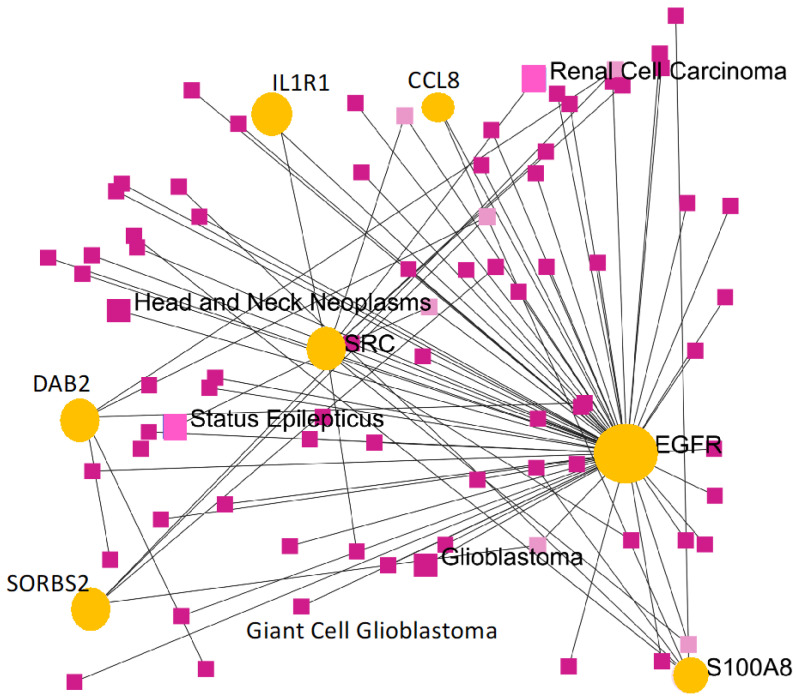
A gene-disease association network of the most common hub genes of co-expression.

**Table 1 genes-13-00518-t001:** The number of down and up-regulated DEGs by paired features.

Treatments Compared	Down-Regulated DEGs	Up-Regulated DEGs
GBT–GSN	592	981
GBT–GSN_H	728	1192
GBT–GSH	562	894
GBT–GSH_N	448	867
GSN-GSN_H	30	44
GSN-GSH	40	5
GSN-GSH_N	4	3
GSN_H-GSH	233	301
GSN_H-GSH_N	0	0
GSH-GSH_N	324	242

**Table 2 genes-13-00518-t002:** The GO and KEGG pathway enrichments of the 61 common hub genes in the turquoise module.

Category	Term	Count	%	*p*-Value
GOTERM_BP_FAT	GO:0006955~immune response	199	20.7507821	5.93 × 10^−36^
	GO:0006952~defense response	194	20.2294056	1.00 × 10^−34^
	GO:0050900~leukocyte migration	83	8.6548488	4.11 × 10^−30^
	GO:0006954~inflammatory response	109	11.3660063	2.26 × 10^−29^
	GO:0002682~regulation of immune system process	170	17.7267988	2.77 × 10^−28^
	GO:0016477~cell migration	150	15.641293	5.56 × 10^−26^
	GO:0051674~localization of cell	160	16.6840459	2.47 × 10^−25^
	GO:0048870~cell motility	160	16.6840459	2.47 × 10^−25^
	GO:0040011~locomotion	173	18.0396246	1.99 × 10^−24^
	GO:0002684~positive regulation of immune system process	128	13.3472367	4.78 × 10^−24^
GOTERM_CC_FAT	GO:0005887~integral component of plasma membrane	156	16.2669447	7.49 × 10^−13^
	GO:0031226~intrinsic component of plasma membrane	159	16.5797706	2.26 × 10^−12^
	GO:0005615~extracellular space	141	14.7028154	2.42 × 10^−12^
	GO:0044421~extracellular region part	291	30.3441085	7.48 × 10^−11^
	GO:0031988~membrane-bounded vesicle	275	28.6757039	7.59 × 10^−11^
	GO:0045121~membrane raft	45	4.6923879	1.03 × 10^−10^
	GO:0098857~membrane microdomain	45	4.6923879	1.16 × 10^−10^
	GO:0009986~cell surface	83	8.6548488	1.65 × 10^−09^
	GO:0098589~membrane region	49	5.10948905	3.57 × 10^−09^
	GO:0005576~extracellular region	326	33.9937435	4.78 × 10^−09^
GOTERM_MF_FAT	GO:0005102~receptor binding	128	13.3472367	6.62 × 10^−09^
	GO:0032403~protein complex binding	79	8.23774765	2.52 × 10^−09^
	GO:0005178~integrin binding	23	2.3983316	1.31 × 10^−08^
	GO:0004872~receptor activity	136	14.181439	2.61 × 10^−08^
	GO:0060089~molecular transducer activity	136	14.181439	2.61 × 10^−08^
	GO:0003779~actin binding	47	4.90093848	1.39 × 10^−07^
	GO:0005539~glycosaminoglycan binding	30	3.1282586	7.18 × 10^−07^
	GO:0004871~signal transducer activity	135	14.0771637	7.64 × 10^−07^
	GO:0098772~molecular function regulator	109	11.3660063	5.07 × 10^−06^
	GO:0038023~signaling receptor activity	110	11.4702815	5.65 × 10^−06^
KEGG_PATHWAY	hsa05150:Staphylococcus aureus infection	20	2.08550574	7.31 × 10^−11^
	hsa05144:Malaria	19	1.98123045	1.02 × 10^−10^
	hsa04380:Osteoclast differentiation	29	3.02398332	1.38 × 10^−09^
	hsa04064:NF-kappa B signaling pathway	22	2.29405631	1.73 × 10^−08^
	hsa05134:Legionellosis	17	1.77267988	4.00 × 10^−08^
	hsa04610:Complement and coagulation cascades	18	1.87695516	2.97 × 10^−07^
	hsa05133:Pertussis	18	1.87695516	1.06 × 10^−06^
	hsa04015:Rap1 signaling pathway	32	3.33680918	1.48 × 10^−06^
	hsa04611:Platelet activation	23	2.3983316	5.55 × 10^−06^
	hsa04060:Cytokine-cytokine receptor interaction	33	3.44108446	1.18 × 10^−05^

**Table 3 genes-13-00518-t003:** The most common hub genes of co-expression and PPI networks of GBM. Gene expression and Fold Change (FC) values are converted log_2_ base, and Betweenness Centrality (BC).

Gene ID	Genes	Nodes	BC	Expression	FC
6714	*SRC*	30	3695.91	2.6267	2.9203
6850	*SYK*	16	1439.37	3.4004	2.7203
1956	*EGFR*	15	1418.53	3.5177	2.9542
4067	*LYN*	14	583.26	3.4102	3.0925
5880	*RAC2*	13	1008.07	3.1154	3.1027
25663	*IL1R1*	11	559.3	1.7369	3.2063
8470	*SORBS2*	11	628.4	1.4228	2.7049
5336	*PLCG2*	11	559.3	1.9957	3.2063
6279	*S100A8*	10	331.58	1.5018	3.4303
20307	*CCL8*	10	594.37	1.6424	2.7071
5294	*PIK3CG*	9	578.13	3.0209	2.5033
391	*RHOG*	9	289.17	2.9575	3.1110
960	*CD44*	8	928.38	3.4897	2.7970
7099	*TLR4*	8	541.23	1.9957	3.2063
58480	*RHOU*	8	84.23	3.0484	3.1619
1958	*EGR1*	7	851.81	1.5607	2.3868
13132	*DAB2*	7	538.61	1.2431	3.2141
3791	*KDR*	7	255.53	3.2197	2.4276
3689	*ITGB2*	6	754.92	3.1132	2.5583
3055	*HCK*	6	753.11	3.4433	2.7049

**Table 4 genes-13-00518-t004:** The KEGG pathways of the top 20 candidate hub genes of PPI network in GBM gene expression dataset.

Term	%	*p*-Value	Genes	FDR
hsa04370:VEGF signaling pathway	30	9.46 × 10^−7^	*LYN*, *HCK*, *SYK*, *PLCG2*, *RAC2*, *PIK3CG*	7.29 × 10^−5^
hsa05205:Proteoglycans in cancer	35	3.52 × 10^−6^	*SRC*, *PLCG2*, *KDR*, *TLR4*, *CD44*, *EGFR*, *PIK3CG*	1.35 × 10^−4^
hsa04666:Fc gamma R-mediated phagocytosis	25	9.41 × 10^−6^	*SRC*, *PLCG2*, *RAC2*, *KDR*, *PIK3CG*	2.37 × 10^−4^
hsa04664:Fc epsilon RI signaling pathway	25	1.45 × 10^−5^	*LYN*, *SYK*, *PLCG2*, *RAC2*, *PIK3CG*	2.37 × 10^−4^
hsa04662:B cell receptor signaling pathway	25	1.54 × 10^−5^	*LYN*, *SYK*, *PLCG2*, *RAC2*, *PIK3CG*	2.37 × 10^−4^
hsa04064:NF-kappa B signaling pathway	25	3.87 × 10^−5^	*LYN*, *SYK*, *IL1R1*, *PLCG2*, *TLR4*	4.96 × 10^−4^
hsa04062:Chemokine signaling pathway	30	4.69 × 10^−5^	*LYN*, *HCK*, *CCL8*, *SRC*, *RAC2*, *PIK3CG*	5.16 × 10^−4^
hsa04015:Rap1 signaling pathway	30	8.39 × 10^−5^	*SRC*, *ITGB2*, *RAC2*, *KDR*, *EGFR*, *PIK3CG*	8.07 × 10^−4^
hsa04650:Natural killer cell mediated cytotoxicity	25	1.45 × 10^−4^	*SYK*, *ITGB2*, *PLCG2*, *RAC2*, *PIK3CG*	0.00111904
hsa05169:Epstein-Barr virus infection	25	1.45 × 10^−4^	*LYN*, *SYK*, *PLCG2*, *CD44*, *PIK3CG*	0.00111904
hsa04611:Platelet activation	25	1.86 × 10^−4^	*LYN*, *SYK*, *SRC*, *PLCG2*, *PIK3CG*	0.00130087
hsa05120:Epithelial cell signaling in Helicobacter pylori infection	20	4.52 × 10^−4^	*LYN*, *SRC*, *PLCG2*, *EGFR*	0.00289853
hsa04012:ErbB signaling pathway	20	9.71 × 10^−4^	*SRC*, *PLCG2*, *EGFR*, *PIK3CG*	0.00575409
hsa04510:Focal adhesion	25	0.0010709	*SRC*, *RAC2*, *KDR*, *EGFR*, *PIK3CG*	0.00589005

**Table 5 genes-13-00518-t005:** The five most significant hub genes by average expression values in log_2_ base for each treatment.

Treatments					
Hub Genes	GBT	GSN	GSN_H	GSH	GSH_N
*IL1R1*	2.14414492	1.51946703	1.62000278	1.69623124	1.70512233
*SORBS2*	1.53735033	1.3820224	1.40838817	1.41361128	1.40112064
*S100A8*	3.02782655	1.16171985	1.11443605	1.07375792	1.13145547
*CCL8*	2.15629112	1.5202201	1.46912071	1.53510117	1.53155342
*DAB2*	1.33510619	1.18921123	1.29003408	1.21577006	1.18538083

**Table 6 genes-13-00518-t006:** Base-2 logarithmic scale of differential expression of most significant hub genes in two different GBM datasets.

Datasets	Genes	Expression	FC	*p*-Value
GSE45117	*IL1R1*	1.73699366	3.2063	1.40 × 10^−11^
	*SORBS2*	1.42849856	2.7049	2.45 × 10^−9^
	*S100A8*	1.50183917	3.4303	3.95 × 10^−10^
	*CCL8*	1.6424573	2.7071	3.65 × 10^−11^
	*DAB2*	1.24310048	3.2141	1.23 × 10^−8^
GSE124145	*IL1R1*	1.85324517	1.5956	1.76 × 10^−5^
	*SORBS2*	2.04573421	2.4535	1.95 × 10^−3^
	*S100A8*	3.47524211	5.6832	1.44 × 10^−6^
	*CCL8*	1.89256437	3.7544	1.47 × 10^−7^
	*DAB2*	3.56778915	4.8697	3.56 × 10^−5^

## Data Availability

The GSE45117 dataset used and analyzed in this present study are available in the NIH GEO (http://www.ncbi.nlm.nih.gov/geo, accessed on 1 February 2022) public repository.

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
