# Peer review of "Screening the Significant Hub Genes by Comparing Tumor Cells, Normoxic and Hypoxic Glioblastoma Stem-like Cell Lines Using Co-Expression Analysis in Glioblastoma"

_genes, 2022, doi:10.3390/genes13030518_

Round 1
Reviewer 1 Report
- Line 65, 'This study aims to identify that affect cells and suggest a treatment to handle them to block the rapid progress' - unclear statement.
- Would authors be able to confirm the expression IL1R1, SORBS2, S100A8, CCL8, and DAB2 using PCR?
- Using one dataset may not be sufficient, I suggest authors use and compare with other datasets and include the clinical dataset too.
- The hub genes of IL1R1, SORBS2, S100A8, CCL8, and DAB2 - to be used in GBM as markers or therapeutic targets are still unclear? In fact, using different datasets with the relevant mutational status of clinical samples would provide better insights. Also, prediction or pathway analysis of the fives genes in GBM mechanisms - progression/therapy would be interesting and add value. The current survival analysis and GO are still general/vague.
Author Response
Reviewer 1 (Anonymous)
Comment 1) Line 65, 'This study aims to identify that affect cells and suggest a treatment to handle them to block the rapid progress' - unclear statement.
Response 1) Thanks a lot for this comment. It is now edited and highlighted with yellow in Line 65.
Comment 2) Would authors be able to confirm the expression IL1R1, SORBS2, S100A8, CCL8, and DAB2 using PCR?
Response 2) It would be great to confirm gene targets using PCR. However, it requires wet lab of GBM tumor samples (from real patients or other species). The study is designed based on using bioinformatics and computational tools. We utilized one of the best approaches available in hand. Another approach might be using non-negative matrix factorization (NMF). We could definitely add this comment as a limitation of the study and other approaches in the discussion section. Please refer to the discussion section with yellow highlighted text.
Comment 3) Using one dataset may not be sufficient, I suggest authors use and compare with other datasets and include the clinical dataset too.
Response 3) Thanks for pointing this out. We have added GSE124145 dataset which is a RNAseq microarray gene expression dataset of glioma and glioma stem cells to see how much of the DEGs are in common of the GSE45117 and GSE124145 in Lines 149-158.
Comment 4) The hub genes of IL1R1, SORBS2, S100A8, CCL8, and DAB2 - to be used in GBM as markers or therapeutic targets are still unclear? In fact, using different datasets with the relevant mutational status of clinical samples would provide better insights. Also, prediction or pathway analysis of the five genes in GBM mechanisms - progression/therapy would be interesting and add value. The current survival analysis and GO are still general/vague.
Response: Thank you so much. We have added another similar clinical data set GSE124145. And saw that our real hub genes were also upregulated in GSE124145. And those new texts along with Table 6 are added in Lines 299-306.
We have constructed a gene-disease association network using DIsGeNET database on common hub genes to confirm our results are valid. Please see the highlighted text in Lines 160-165. We had the survival analysis results through UALCAN using TCGA database of GBM tissues compared with normal brain tissues and the results come out at a significance level p-value<0.05 which shows promising results for this in-silico study.
Reviewer 2 Report
To editors and authors
This is an interesting paper focusing on "glioblastoma and glioma stem-like cells". This paper is corresponding to Genes aim and scope. Thus, I recommend this manuscript for publication after some revisions below.
1) Please check again author guideline and some recent published articles, some part showed that citation and reference format did not follow MDPI.
2) Please add statistical part.
3) Please add limitation part and further direction of research.
Sincerely
Author Response
Reviewer 2 (Anonymous)
This is an interesting paper focusing on "glioblastoma and glioma stem-like cells". This paper is corresponding to Genes aim and scope. Thus, I recommend this manuscript for publication after some revisions below.
Response: Thanks a lot for this comment. We have taken all of your suggestions into consideration as the following,
Comment 1) Please check again author guideline and some recent published articles, some part showed that citation and reference format did not follow MDPI.
Response 1) Thanks a lot for this comment. We have definitely followed the format of the MDPI in the revised version. Please refer to the main text and references.
Comment 2) Please add statistical part.
Response 2) Please, section 2.3 and the yellow highlighted text in Section 2.3.
Comment 3) Please add limitation part and further direction of research.
Response 3) Please, see Lines 420-425.
Reviewer 3 Report
The manuscript entitled “Identification of hub genes between glioblastoma and glioma stem-like cells using co-expression analysis” is a well-written manuscript. The authors retrieved the gene expression dataset from GEO and analyzed using R packages. Here are my recommendations to improve the quality of the manuscript.
- The authors have retrieved the datasets from GEO, have they also checked the GB datasets from the SRA database? If yes, then authors should report the SRA IDs as well.
- In figure 7. HR ratio is missing in survival analysis. Authors’ should also include them.
Author Response
The manuscript entitled “Identification of hub genes between glioblastoma and glioma stem-like cells using co-expression analysis” is a well-written manuscript. The authors retrieved the gene expression dataset from GEO and analyzed using R packages. Here are my recommendations to improve the quality of the manuscript.
Comment 1) The authors have retrieved the datasets from GEO, have they also checked the GB datasets from the SRA database? If yes, then authors should report the SRA IDs as well.
Response 1) This is a very nice recommendation but we already added a gene-disease information network utilizing hub genes and we have identified another GBM data set and reported the results in the revised versions.
Comment 2) In figure 7. HR ratio is missing in survival analysis. Authors’ should also include them.
Response 2) Sorry about that part. We would of course add them. Thanks a lot for this comment. Please, see the p(HR) values on the updated plots of Figure 7.